# TWIN-SCAN: Liver Biomarker Estimation Using Machine Learning and Digital Twin Simulation

Sumaiya Afroz Mila, Sandip Ray
Department of ECE, University of Florida, Gainesville, FL 32611. USA.
mila.s@ufl.edu, sandip@ece.ufl.edu

*Abstract*—Early-stage liver diseases often progress silently and remain undiagnosed due to the lack of labeled clinical biomarker data. This study presents a comparative modeling framework to predict stage-specific liver biomarker ranges using limited real-world data. We compare two complementary approaches: (1) a machine learning (ML) based mapping method trained on cirrhosis-stage data and healthy references, and (2) a physiology-informed partial digital twin (DT) model that simulates bilirubin conjugation across various stages of liver disease. The digital twin is further personalized using patient-level parameters such as age and gender to enhance accuracy. Results demonstrate that the digital twin approach yields biomarker predictions that are more biologically plausible and better aligned with clinical trends. This framework highlights the potential of physiological modeling to complement data-driven methods in generating accurate, non-invasive biomarker estimates for early-stage liver disease detection during data scarcity.

*Index Terms*—Early-Stage Liver Disease, Biomarker Estimation, Digital Twin Modeling, Bilirubin Conjugation

## I. INTRODUCTION

Liver disease is currently one of the most significant and rapidly growing global health challenges. It often progresses silently through several stages, from nonalcoholic fatty liver disease (NAFLD) to fibrosis to cirrhosis, and in many cases, to hepatocellular carcinoma (HCC). Despite being a vital organ responsible for various vital functions, such as-detoxification, metabolism, bile production, and protein synthesis, liver lacks nerve endings, meaning early liver damage presents no symptoms. By the time liver dysfunction becomes clinically evident (typically at the cirrhosis stage) the disease is already advanced, and treatment options shift from prevention or reversal to long-term management.

The global issue of liver cirrhosis continues to rise. In 2017, there were approximately 520,000 new cases of cirrhosis and chronic liver disease. By 2019, cirrhosis had caused 1.48 million deaths worldwide, an 8.1% increase from 2017 [1]. While cirrhosis from hepatitis B or C is declining due to effective antiviral interventions, cirrhosis caused by alcohol and NAFLD is increasing rapidly. In the United States alone, cirrhosis affects about 0.27% of the adult population, equivalent to 633,323 people every year, and 69% of those individuals are unaware of their condition [2]. While NAFLD is commonly associated with obesity, over 10% of individuals with NAFLD in the U.S. are lean, highlighting the need for non-invasive, population-wide screening approaches [3].

These statistics underscore the urgent need for accessible, non-invasive liver health monitoring solutions, especially for detecting early-stage liver disease. Intervening at earlier stages like NAFLD or fibrosis is critical, as these are the stages where curative interventions are possible through lifestyle changes and proper clinical management. In contrast, cirrhosis represents irreversible liver damage where the focus shifts toward symptom control and complication prevention.

Artificial intelligence (AI) and machine learning (ML) approaches have emerged as powerful tools for early disease detection, including liver-related disorders. However, like any data-driven solution, their effectiveness highly depends on the availability and completeness of training datasets. Public liver disease datasets often focus on diagnosed cirrhosis stages (e.g., cirrhosis stages 1–3), while intermediate stages, particularly NAFLD and early fibrosis remain underrepresented or absent. Digital twin (DT) modeling has been explored in liver-related applications, such as pharmacokinetics, regenerative therapy planning, and surgical guidance. However, there has been limited work on developing DT models for early-stage disease tracking and biomarker-level physiological simulation, particularly for conditions like NAFLD and fibrosis. A key motivation behind this work is the need to evaluate modeling feasibility for early-stage liver disease detection in data-scarce conditions. This gap underlies our focus on biomarker estimation using physiology-informed DT modeling.

This paper addresses a critical gap in liver disease research: the lack of systematic comparisons between the biomarker estimation performance of ML-based and physiology-informed approaches in early-stage conditions such as NAFLD and fibrosis. While existing studies have explored each approach independently, to the best of our knowledge, no existing study has directly evaluated ML-based biomarker mapping alongside physiology-driven simulations under a unified framework. To address this, we present a comparative modeling framework that evaluates the performance of both methods for estimating bilirubin levels in data-limited scenarios, to support reliable early-stage liver disease prediction solutions, disease progression monitoring and treatment efffect tracking. Our contribution include:

- **Regression-based ML mapping method** trained on cirrhosis-stage [4] and healthy reference data, designed to extrapolate likely biomarker ranges for underrepresented stages while highlighting the limitations of extrapolating under sparse labels.
- **Physiology-informed digital twin model**, TWIN-SCAN (A Digital **Twin** Approach to **S**imulating Bilirubin

**C**onjug**a**tio**n**), that simulates bilirubin conjugation, enabling biologically grounded and interpretable biomarker estimation across disease stages; and

- **Three TWIN-SCAN variants:** (i) a Generic model using disease stage alone as input, (ii) a Tier-1 Personalized model incorporating gender-specific physiological parameters, and (iii) a Tier-2 Personalized model adding gender and age-specific adjustments.

We evaluate the predictive performance of these models using a consistent data and metric. The results show that while the ML-based method provides broad estimation ranges, TWIN-SCAN yields more interpretable and physiologically grounded predictions. Together, these contributions support structured comparison of data-driven and physiology-based approaches and lay the foundation for building interpretable, and non-invasive, early-stage liver disease monitoring solutions using limited real-world data.

## II. RELATED WORK

Liver disease progresses through a series of stages, beginning with hepatic fat accumulation (NAFLD), followed by inflammation, fibrosis, tissue scarring, and ultimately cirrhosis and hepatocellular carcinoma. Since liver lacks sensory nerves, early-stage disease is often asymptomatic, leading to delayed diagnosis.

### A. Current Research on Liver Disease Detection

Early detection of NAFLD is crucial because it is a reversible condition with lifestyle interventions, dietary modifications, and proper medication. In contrast, once liver damage progresses to late stage fibrosis or early cirrhosis, medical interventions become expensive and largely focus on disease management rather than cure. Traditional diagnostic methods, such as liver biopsy, are typically reserved for advanced stages due to their invasive nature, high cost, and associated risks [5]. Non-invasive imaging techniques such as ultrasound, computed tomography (CT), magnetic resonance imaging (MRI), and X-ray have been developed to address these limitations, with magnetic resonance elastography (MRE) and ultrasound-based elastography being commonly used for fibrosis assessment [6]. Magnetic resonance imaging-proton density fat fraction (MRI-PDFF) is used for detecting hepatic steatosis [5], [7], [8]. Despite these advances, *imaging is often reserved for high-risk or symptomatic patients, and applied inconsistently even in these cases, limiting its effectiveness in early detection, preventive or curative interventions and continuous liver health monitoring. [9], [10].*

### B. Current Research on NAFLD Detection

Current research for NAFLD detection is primarily focused on non-invasive imaging techniques such as magnetic resonance imaging-proton density fat fraction (MRI-PDFF) and transient elastography (TE) to measure liver fat [7], [11], [12]. Studies have demonstrated that MRI-PDFF provides strong correlations with histologic liver fat measurements and outperforms traditional ultrasound or CAP methods in detecting hepatic steatosis [11], [12].

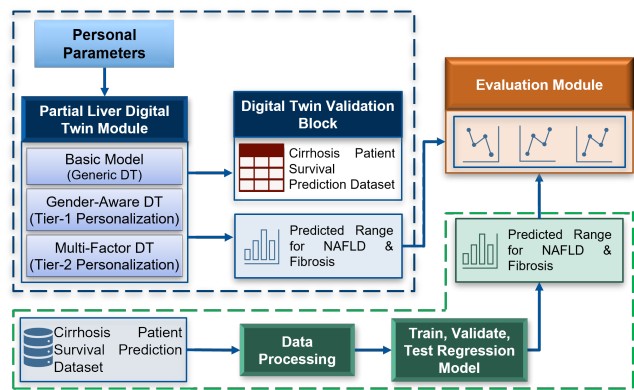

Fig. 1. System architecture of the comparative modeling framework.

However, the detection of early NAFLD remains challenging. Most studies rely on indirect extrapolation from fibrosis or cirrhosis stages rather than direct identification of early NAFLD [5]. Imaging studies predominantly validate liver fat quantification against biopsy-confirmed steatosis rather than detecting true NAFLD directly. Additionally, liver biopsy itself is invasive, costly, and not part of routine clinical screening, further limiting data to selected high-risk populations. Thus, while imaging biomarkers have improved liver fat measurement, early-stage NAFLD detection remains limited and heavily reliant on inferred correlations rather than direct diagnosis.

### C. Liver Digital Twin Models

Digital Twin (DT) models simulate organ-level physiology in real time and allow patient-specific predictions of health trajectories. In hepatology, DTs can integrate patient data such as age, gender, biomarkers, and lifestyle indicators to create virtual liver profiles that dynamically evolve with disease progression or treatment [13]. These models support minimally invasive, continuous monitoring and preventive care by identifying early deviations from healthy trends often before critical thresholds are crossed.

Most liver-focused DT models that have been developed are designed for pharmacokinetic modeling, regenerative therapy planning, or surgical guidance. For example, virtual liver platforms have been developed to evaluate drug-induced liver injury [14], predict post-hepatectomy outcomes [15], and model liver regeneration at the cellular orchestration level [16]. Some works have integrated DTs with imaging-based procedural guidance [17], [18]. Current advancements, though valuable, still fall short in addressing the need for a physiologically grounded, minimally invasive digital twin system that enables real-time tracking of biomarker trends and early detection of NAFLD progression.

## III. METHODOLOGY

Our work addresses the gaps highlighted in the previous section by developing TWIN-SCAN, a physiology-informed model of bilirubin conjugation, and evaluating its predictive performance alongside an ML-based biomarker mapping

TABLE I
BILIRUBIN DISTRIBUTION BY CIRRHOSIS STAGE IN CIRRHOSIS PATIENT SURVIVAL PREDICTION DATASET

| Cirrhosis Stage | Total Samples | Mean | Std Dev | Min | Max | 12.5%–87.5% Range |
|---|---|---|---|---|---|---|
| Stage 1 | 8,265 | 2.47 | 3.79 | 0.3 | 28.0 | [0.5 − 5.4] |
| Stage 2 | 8,441 | 3.32 | 4.88 | 0.3 | 28.0 | [0.7 − 7.2] |
| Stage 3 | 8,294 | 4.42 | 5.14 | 0.3 | 28.0 | [0.8 − 11.4] |

method. To support liver disease detection at an early-stage through biomarker analysis, we propose a comparative modeling framework designed to estimate biomarker ranges for the underrepresented stages. We specifically focus on bilirubin as the target biomarker due to its strong clinical relevance in liver dysfunction and its dominant contribution observed in our preliminary principal component analysis (PCA), which is discussed in detail in Section IV. As illustrated in Fig. 1, this framework develops and evaluates two distinct approaches: (1) ML method that trains various regression models on cirrhosis-stage data to extrapolate biomarker ranges for earlier stages, and (2) TWIN-SCAN, a partial digital twin model of the liver that simulates disease-stage-specific bilirubin values by modeling bilirubin conjugation pathways. Personalization tiers in TWIN-SCAN incorporate patient-level parameters (e.g., gender, age) to improve the digital twin's estimation accuracy. We evaluate the performance of both approaches and compare their predictive capabilities against real-world clinical data to assess their effectiveness in estimating biomarkers for early-stage liver disease detection.

### A. Biomarker Selection Rationale using PCA:

To guide the selection of a target biomarker, we have applied Principal Component Analysis (PCA) as a feature selection method to identify those biomarkers that account for the highest variance across patient samples. Since features contributing most to the top principal components are likely to have greater relevance, and bilirubin consistently emerged as the top contributor to the first principal component (PC1), we select it as the key variable for our method. Clinically, elevated bilirubin is widely recognized as an indicator of liver dysfunction [19]. Therefore, integrating PCA with domain knowledge allowed us to both statistically and clinically justify our focus on bilirubin in the TWIN-SCAN model.

### B. Machine Learning Approach for Estimating Early-Stage Biomarker Ranges

The ML workflow depicted in Fig. 1 (highlighted by the green dashed box) outlines the steps we employ to estimate the biomarker ranges at early stages of liver disease, where labeled clinical data is often missing. We utilize available data from healthy individuals and cirrhosis-stage patients to infer likely bilirubin ranges for intermediate stages.

*Dataset Description:* We utilize the Cirrhosis Patient Survival Prediction dataset, which contains patient-level biomarker values (e.g., bilirubin) and corresponding cirrhosis stage labels (Stages 1–3). After filtering out the records with missing bilirubin values, we have a total of 25,000 samples,

nearly evenly distributed across all three cirrhosis stages: Stage 1 (8,265), Stage 2 (8,441), and Stage 3 (8,294).

The bilirubin distribution increases with disease severity in both mean and spread. Table I summarizes the total sample count, mean, standard deviation, and high-density range (defined as the 12.5th to 87.5th percentile) for each cirrhosis stage. Stage 3 shows the widest variability, reflecting clinical heterogeneity in advanced disease. Since this dataset lacks entries for earlier liver disease stages such as NAFLD and fibrosis, we integrate 8000 synthetic healthy bilirubin ranges (0.1–1.3mg/dL) derived from population-level statistical studies to it [20], [21]. The dataset is then processed so that class 0 corresponds to healthy, classes 3–5 represent cirrhosis stages 1–3, and the underrepresented stages NAFLD (class 1) and fibrosis (class 2) are reserved for prediction.

*Regression Model Development:* Various regression models[1] such as, linear regression, support vector regression (SVR), XGBoost, random forest regression are then trained on the processed dataset, where healthy reference values and cirrhosis stages 1, 2, and 3 are used in different combinations depending on the target stage for bilirubin estimation. Once trained, the models are used to estimate the likely bilirubin range for class 3 (cirrhosis stage 1), class 1 (NAFLD) and class 2 (fibrosis). This ML-based extrapolation provides a statistically grounded yet somewhat interpretable estimate of missing biomarker stages. Although NAFLD and fibrosis occur between healthy and cirrhosis stages in disease progression (interpolation task in clinical term), from a modeling standpoint, the ML model performs extrapolation since it is trained only on the endpoints (healthy and cirrhosis) and lacks direct examples of these intermediate stages.

### C. Development of TWIN-SCAN: Physiology-Driven Partial Liver Digital Twin

As a physiology-informed alternative to the ML-based approach, we design TWIN-SCAN, a partial digital twin (DT) of the liver focusing on bilirubin conjugation pathway, shown in Fig. 1(highlighted by the blue dashed box). In the human body, unconjugated bilirubin is produced as a waste byproduct from the breakdown of heme in red blood cells. This process involves several enzymatic steps: macrophage-mediated hemolysis releases heme, which is then converted to biliverdin by heme oxygenase and subsequently to unconjugated bilirubin by biliverdin reductase. The liver then conjugates bilirubin using the enzyme UDP-glucuronyl transferase, making it water-

---

[1]Regression is used instead of linear interpolation to account for non-uniform, stage-dependent biomarker changes and to improve predictive robustness in data-scarce regions.

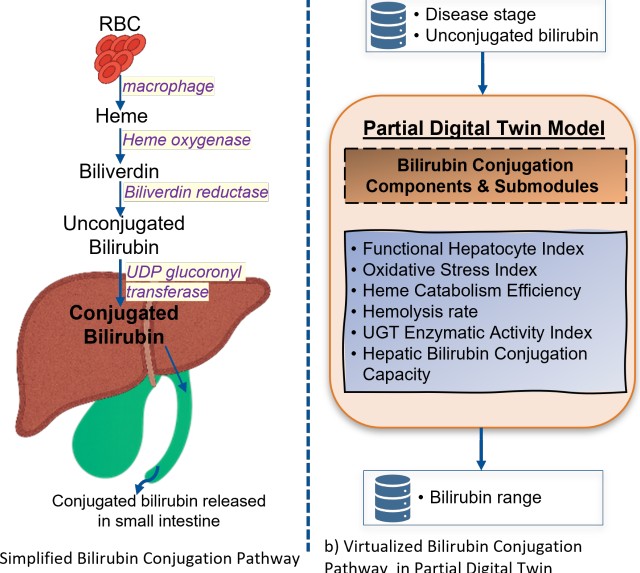

**a) Simplified Bilirubin Conjugation Pathway**

**b) Virtualized Bilirubin Conjugation Pathway in Partial Digital Twin**

Fig. 2. Bilirubin Conjugation: Real-World Physiology and Digital Twin Representation.

---

**Algorithm 1** Digital Twin Estimation of Unconjugated Bilirubin

**Input:** Disease Stage $D \in \{0, 1, 2, 3, 4, 5\}$
        Age $A \in \mathbb{R}^+$ (Personalized DT only)
        Gender $G \in \{\text{Male}, \text{Female}\}$ (Personalized DT only)

**Output:** Estimated unconjugated bilirubin range `unconj_bilirubin` (mg/dL)

1: $f_{\text{hep}} \leftarrow f_1(D)$       // Functional hepatocyte fraction
2: `ox_stress` $\leftarrow f_2(D)$       // Oxidative stress
3: `spread` $\leftarrow f_3(D)$       // Disease spread level
4: `catabolism_eff` $\leftarrow f_4(\text{spread}, A)$     // Heme catabolism efficiency
5: `hemolysis` $\leftarrow f_5(\text{spread}, A)$     // Hemolysis rate
6: `UGT` $\leftarrow f_6(\text{spread}, A, G)$     // UGT enzyme activity
7: `conj_capacity` $\leftarrow f_7(\text{spread}, A, G)$     // Bilirubin conjugation capacity
8:
9: `unconj_bilirubin` $\leftarrow \mathcal{F}(f_{\text{hep}}, \text{ox\_stress}, \text{UGT},$
10:     $\text{hemolysis}, \text{conj\_capacity}, \text{catabolism\_eff})$
11: **return** `unconj_bilirubin`

---

soluble for excretion via bile. A simplified illustration of this physiological pathway is shown in Fig. 2*a*.

This biological process is translated into virtual space by designing TWIN-SCAN through a set of submodules: Functional Hepatocyte Index, Oxidative Stress Index, Heme Catabolism Efficiency, Hemolysis Rate, UGT Enzymatic Activity Index, and the resulting Hepatic Bilirubin Conjugation Capacity. Fig. 2*b* shows the major modules of TWIN-SCAN in virtual space and Algorithm 1 describes the sequential computation steps followed in its implementation. For each patient, the estimated bilirubin range is computed following this method, and the summary of TWIN-SCAN range shown in the result section is calculated by aggregating the low and high endpoints across all patients. TWIN-SCAN is developed in two configurations:

- **Generic Model:** This base model of TWIN-SCAN uses only disease stage as input. Each submodule simulates how physiological impairments across *disease stages* affect bilirubin production and excretion. Using disease state as input, the model simulates a plausible range for bilirubin value based on these parameters, producing biologically interpretable outputs for stages absent in the dataset.
- **Personalized Model (Tier-1 and Tier-2):** To improve biomarker estimation range, the TWIN-SCAN framework is extended with patient-level biological parameters. In the first tier, a Gender-Aware model adjusts key physiological coefficients based on known sex-specific differences in liver metabolism. In the second tier, a Multi-Factor model incorporates both gender and age, allowing more precise adjustments to parameters such as UGT enzymatic activity and functional hepatocyte [22].

### D. *Performance Evaluation of* TWIN-SCAN *and ML-based Bilirubin Estimation*

Estimated bilirubin ranges from each variant (Generic, Tier-1, Tier-2) of TWIN-SCAN are first evaluated using patient-level bilirubin values from cirrhosis stages in the clinical dataset. Estimations from the regression-based ML model for cirrhosis stage 1 are compared against corresponding ground truth values. Once validated, both TWIN-SCAN and the ML models are applied to estimate bilirubin ranges for NAFLD and fibrosis stages. We evaluate the performance of each approach as a combination of two key metrics: coverage accuracy, which measures the percentage of real-world bilirubin values captured within the predicted range and mean absolute error, which measures the average deviation between predicted and actual values. Since our digital twin model returns a range rather than a single-point prediction, we compute it as the absolute difference between the midpoint of the DT-estimated range and the mean of the ground truth clinical values. This formulation ensures that the center of the predicted range aligns with the region of highest data density in real-world observations. Evaluating model performance based on coverage alone can be misleading, as estimating a wide bilirubin range, from a physiologically low to a high extreme would always yield 100% coverage accuracy. Therefore, the combination of coverage accuracy and midpoint-based MAE provides a more robust and clinically meaningful evaluation, capturing true values without producing implausibly wide intervals. The combination of these metrics ensures that model results are both reliable and clinically meaningful, capturing real values without producing overly broad or implausible prediction intervals. By evaluating the performance of all models (TWIN-SCAN and ML mapping) against each other and the clinical dataset, we validate the effectiveness of this comparative modeling

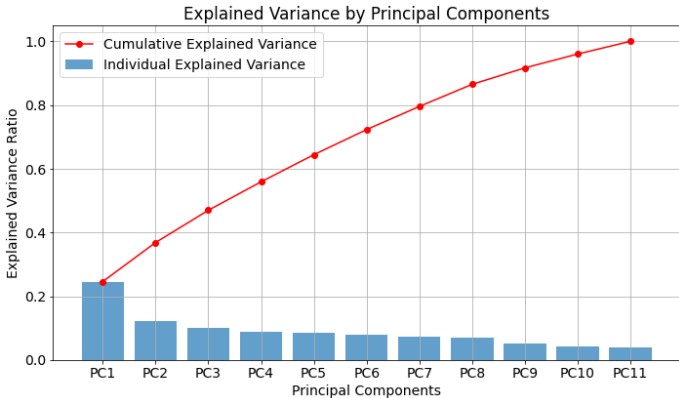

Fig. 3. PCA Scree Plot of Liver Biomarker Features.

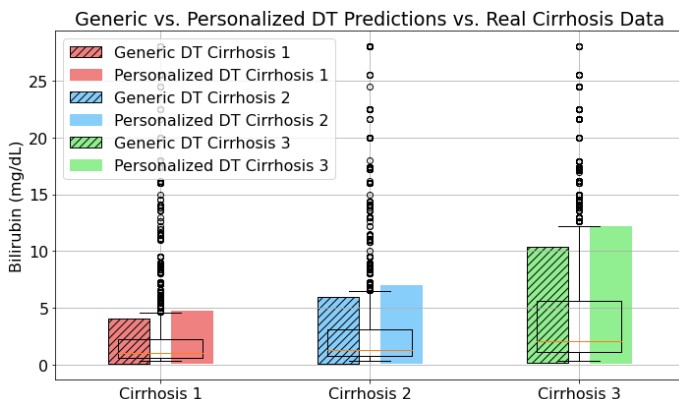

Fig. 4. Predicted bilirubin range by generic DT and personalized DT model with ground truth value.

framework in generating interpretable, physiology-driven predictions. These findings support its potential utility in non-invasive, early-stage liver disease detection, particularly for data-scarce stages.

## IV. EXPERIMENTAL RESULTS

### A. Principal Component Analysis (PCA): Justifying Biomarker Selection

To support our focus on bilirubin, we first conducted Principal Component Analysis (PCA) on the biomarker dataset. As shown in Fig. 3, the first principal component (PC1) captures the highest variance (explained variance ratio = 0.25), with bilirubin emerging as its top contributing feature. This suggests that bilirubin levels account for a substantial portion of inter-patient variability, aligning with its clinical significance as a key indicator of liver dysfunction [19]. These findings support our decision to focus on bilirubin simulation within the digital twin framework.

### B. TWIN-SCAN Validation: Simulated Bilirubin Conjugation vs. Clinical Biomarker Data

We first develop a generic version of TWIN-SCAN which simulates bilirubin conjugation based on disease stage input. As shown in algorithm 1, TWIN-SCAN simulates the physiological production and excretion of bilirubin and outputs a plausible range of bilirubin for each patient based on their cirrhosis stage. The final bilirubin range shown in the result section is summarized by aggregating the low and high endpoints across all patients. As illustrated in Fig. 4, the generic model's results align with the real-world clinical data, modeling the progressive elevation of bilirubin with disease severity.

To improve prediction range and reflect inter-individual physiological differences, we develop personalized variants of TWIN-SCAN by incorporating two patient-specific parameters: age and gender. These variables directly influence hepatic metabolism, hemolysis rate, and UGT enzyme activity, all of which impact bilirubin conjugation capacity. As shown in Fig. 4, the personalized model yields a wider predicted range that has more overlaps with the real-world bilirubin

distributions observed across Cirrhosis stages 1–3, compared to the narrower predictions of the generic model. Incorporating parameters like age and gender in the personalized model improves its sensitivity to real-world deviations, making it better suited for individualized disease monitoring and early intervention.

Table II compares the coverage accuracy (percentage of ground-truth values within predicted range) between the generic and personalized (Tier 2) model. Personalized model improves prediction across all three stages, reaching up to 88.27% for Cirrhosis stage 3. Table III shows mean absolute error (MAE) between the DT-predicted midrange and the actual mean bilirubin levels in the dataset. While both generic and personalized models perform well, the personalized configuration shows lower MAE for cirrhosis stage 1, and stage 2. These results supports our claim that incorporating personal parameters increases physiological resolution and improves overall performance.

### C. ML-Based Biomarker Estimation with Clinical Dataset Validation

Alongside TWIN-SCAN, we develop four regression-based ML models, such as linear regression, support vector regres-

TABLE II
COVERAGE ACCURACY (%) OF DT-PREDICTED BILIRUBIN RANGES

| Stage | Generic DT(%) | Personalized DT(%) |
|---|---|---|
| Cirrhosis 1 | 85.87 | 86.36 |
| Cirrhosis 2 | 85.68 | 86.74 |
| Cirrhosis 3 | 86.52 | 88.27 |

TABLE III
MEAN ABSOLUTE ERROR BETWEEN DT MIDRANGE AND DATASET
MEAN (BILIRUBIN)

| Disease Stage | Generic DT MAE (mg/dL) | Personalized DT MAE (mg/dL) |
|---|---|---|
| Cirrhosis 1 | 0.38 | 0.01 |
| Cirrhosis 2 | 0.27 | 0.26 |
| Cirrhosis 3 | 0.82 | 1.75 |

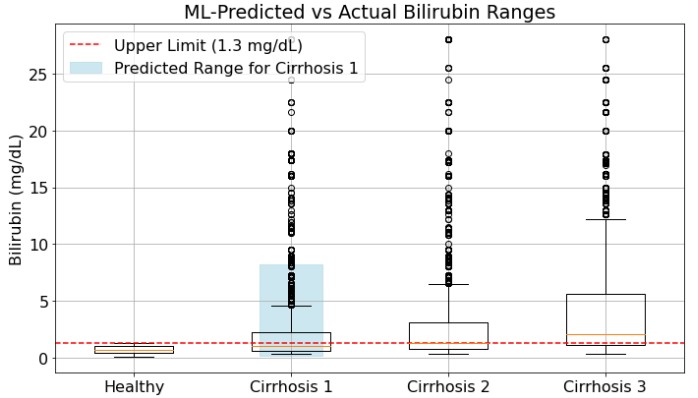

Fig. 5. Regression model's bilirubin prediction for cirrhosis stage 1.

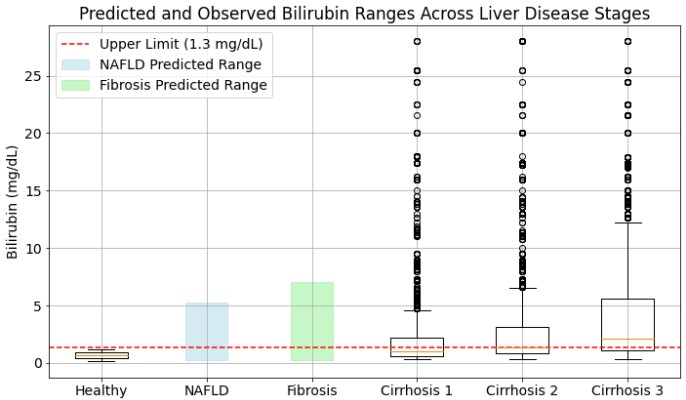

Fig. 6. Regression model's bilirubin prediction for early stage liver disease.

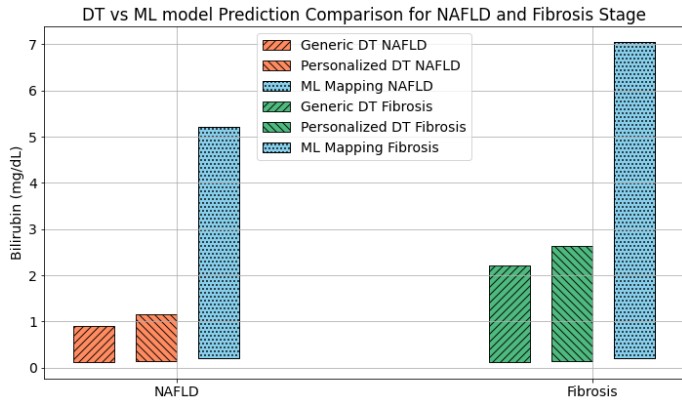

Fig. 7. ML vs DT model's Bilirubin range prediction for NAFLD & Fibrosis stage.

sion (SVR), random forest regression, and gradient boosting regression (XGBoost). These models are first trained on bilirubin values of patients diagnosed with cirrhosis stage 2 and 3, as well as healthy controls; and are then used to estimate values for cirrhosis stage 1. The predicted ranges are validated by comparing them against ground truth values from the Cirrhosis Patient Survival Prediction dataset. Among all the models, XGBoost shows the best overall performance in terms of coverage accuracy and MAE. Therefore, the ML regressor model results reported in this article refer to the XGBoost model. We identify the optimal hyperparameters for XG-Boost through grid-search, and the final configuration of our model is as follows: $n\_estimators = 200, learning\_rate = 0.05, random\_state = 42$. As shown in Fig. 5, the estimated range spans from 0.2 mg/dL to 8.21 mg/dL, with coverage accuracy 91.94% and MAE of 1.73mg/dL, indicating a wider and less physiologically constrained prediction. The trained ML model is then applied to extrapolate likely bilirubin ranges for the underrepresented stages. As shown in Fig. 6, the ML mapping provides a baseline approximation for these early disease stages, reflecting trends learned from extreme stages (healthy and cirrhosis). These ranges align with findings from existing clinical studies that analyze bilirubin trends across fibrosis and advanced stages [19]. Given the limited

availability of labeled clinical data for early-stage liver disease, particularly for underrepresented stages such as NAFLD and fibrosis, the machine learning models adopted in this work serve as practical baselines. The scarcity of stage-specific data limits the effectiveness of more complex or highly tuned models, making simpler regression approaches more appropriate for this extrapolation task. This further motivates the need for physiology-informed frameworks like TWIN-SCAN, which do not rely solely on rich, labeled datasets and can generalize better in data-constrained scenarios.

### D. Comparative Analysis: TWIN-SCAN vs. ML Mapping

While the ML model achieves slightly higher coverage accuracy (91.94%) for cirrhosis stage 1 than TWIN-SCAN (generic-85.87%, personalized-86.36%), it also introduces larger errors (MAE = 1.73mg/dL) due to its reliance on extrapolation and lack of physiological grounding. As shown in Fig. 7, the ML-generated bilirubin ranges for NAFLD and fibrosis are broader and less constrained (blue bars), reflecting its black-box nature.

In contrast, TWIN-SCAN models generate narrower, clinically plausible ranges by modeling the underlying liver function directly and adjusting for patient-specific parameters such as age and gender. This physiology-driven approach improves model performance by decreasing prediction error in data-scarce, early-stage liver disease.

For Cirrhosis Stage 3, the personalized TWIN-SCAN model shows a higher MAE (1.75mg/dL) than the generic variant (0.82mg/dL), likely due to the wider bilirubin range associated with more severe stage, which is approximately twice that of cirrhosis stages 1 and 2. However, it still maintains higher coverage accuracy (88.27%) than generic DT (86.52%) without predicting implausibly wide ranges or extreme outliers. This further highlights the need for evaluating model performance based on coverage accuracy in combination with MAE: while high coverage ensures inclusion of true values, MAE penalizes excessive range width and reflects alignment with the clinical mean.

## V. Conclusion

This work presents a comparative modeling framework and evaluates ML-based biomarker mapping with digital twin modeling to support early-stage liver disease detection. By simulating the physiology of bilirubin conjugation and incorporating patient-specific inputs, the personalized variants of TWIN-SCAN achieve improved bilirubin prediction range over its generic counterpart and shows strong alignment with real-world clinical data. Our results demonstrate that while ML-based methods provide useful estimates, they tend to generate broader and less physiologically grounded ranges with higher MAE, as they rely solely on extrapolated trends from observed data. In contrast, TWIN-SCAN approach produces narrower, clinically plausible predictions by directly modeling liver physiology. Incorporating personal factors such as age and gender enhances the TWIN-SCAN's sensitivity to biological variability, improving prediction accuracy across cirrhosis stages. Notably, this work shows that a partial DT model focused on a single hepatic function (bilirubin conjugation) can provide meaningful insight into disease progression when appropriately personalized.

In future work, we plan to extend TWIN-SCAN into a comprehensive liver model that incorporates non-invasive sensor data (e.g., from wearables or electronic health records), along with physiological pathways (e.g., metabolism, protein synthesis, bile acid production) and a broader set of personalized inputs (e.g., diet, medication, comorbidities). We also plan to validate the model using real-world datasets specifically designed to track biomarker progression in underrepresented stages, allowing the development of personalized digital biomarkers progression tracking solution for proactive liver health monitoring.

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
