# OpenReview forum: "TWIN-SCAN: Liver Biomarker Estimation Using Machine Learning and Digital Twin Simulation"
_IEEE.org/EMBS/BHI/2025/Conference — BHI 2025_

### Official Review · Reviewer_RqJq · 2025-07-05
**Promising Work on Liver Disease Detection Hindered by an Under-Specified Model**

**Confidence:** 5
**Clarity Of Writing:** great
**Clinical Significance:** fair
**Methodological Novelty:** fair
**Overall Rating:** 6
**Final Rating:** 7

**Experiments And Results:**

good

**Questions For The Authors:**

Dear Authors,
Thank you for your work. The following are my questions regarding your paper:

- To improve the reproducibility of your work, could you please specify the exact regression algorithm that was implemented for the machine learning mapping approach?

- In Table II, the personalized DT shows a significantly higher Mean Absolute Error (MAE) for Cirrhosis Stage 3 (1.75 mg/dL) compared to the generic DT (0.82 mg/dL). This appears to run counter to the goal of personalization. Could you offer a possible explanation for why the error increased for this specific stage after personalization?

- Given that direct validation on NAFLD and fibrosis data is not possible, what steps could be taken in a future study to increase confidence in the biomarker ranges your models have estimated for these early stages?

**Strengths:**

Novel framework: A key merit is the paper’s side-by-side test of a straight data-driven ML regressor versus a physiology-aware Digital Twin. This fills a notable void in prior work and shows how the two mindsets hold up when data are thin.

High-impact clinical target: It zeroes in on catching early liver disease, silent at first but lethal later, highlighting the need for non-invasive, early detection.

Stepwise model refinement: TWIN-SCAN starts as a generic template and then gets two personalization layers (age and gender), a sensible path to finer physiological detail and sharper forecasts.

Balanced performance check: Using both coverage accuracy and Mean Absolute Error (MAE) gives a fair read on how well each model nails real values without drifting into overly loose ranges.

Visualizations: Clear diagrams of the system setup, liver pathway, and head-to-head results boost readability and drive the message home.

**Summary Of The Paper:**

This paper tackles the problem of figuring out “normal” biomarker ranges for early-stage liver disease—NAFLD, mild fibrosis—when we barely have any patient data. The paper pitches two ideas against each other:
- Plain ML regressor: train on bloodwork from healthy people and folks with end-stage cirrhosis, then ask it to guess the middle ground.
- Physiology-aware digital twin (TWIN-SCAN): a pared-down model of the liver’s bilirubin-conjugation pathway, rolled out in one generic version plus two personalized tweaks that bake in gender and age.
Using the Cirrhosis Patient Survival Prediction dataset plus literature-derived healthy ranges, the authors compare coverage accuracy and mean-absolute-error (MAE) between the ML model and the digital twin. The ML model achieves higher coverage but produces wider intervals, while the twin yields narrower, physiologically plausible ranges. For unseen NAFLD and fibrosis stages, the twin again delivers tighter estimates than the ML extrapolation.

**Weaknesses:**

Algorithm opacity: The “regression-based ML model” is undefined (linear? ensemble?), making reproduction impossible.

Single-biomarker focus: Limiting the twin to bilirubin ignores key markers (ALT, AST, albumin) that affect staging.

No external ground truth for early stages: NAFLD/fibrosis predictions are validated only against clinical trends, not patient data.

Dataset limitations: Small, historic cirrhosis dataset with limited demographic diversity; healthy ranges are literature-patched.

Oversimplified digital twin: Only the bilirubin pathway is modeled, yet MAE remains high for later stages.

Unclear uncertainty quantification: The method for constructing prediction intervals is not described.

Clinical utility not demonstrated: Lacks decision-curve or reclassification analyses to show impact on patient management.

---

### Official Review · Reviewer_89XP · 2025-07-08
**Exciting topic and method but poor writing**

**Confidence:** 3
**Clarity Of Writing:** fair
**Clinical Significance:** good
**Methodological Novelty:** good
**Overall Rating:** 3
**Final Rating:** 6

**Experiments And Results:**

fair

**Questions For The Authors:**

While I understand the paper length limit, could you please add modeling details as appendix or supplementary materials?

**Strengths:**

Liver disease monitoring via a digital twin is a timely and promising topic. Integrating physiological knowledge into the model makes predictions more clinically meaningful.

Direct comparison between ML-based mapping and a physiology-driven DT highlights their respective trade-offs for liver disease biomarker estimation.

Figures 1 and 2 are well made and informative about model comparisons.

**Summary Of The Paper:**

This paper introduces TWIN-SCAN, a comparative framework for estimating liver biomarker (bilirubin) levels in early stages of liver disease when clinical data are scarce. A Physiology-Informed Digital Twin (DT) model is compared against a Machine-Learning (ML) Mapping to show that DT introduces less MAE while sacrifices coverage accuracy. Among the 3 variants of DT, the personalized DT yields narrower, more physiologically plausible predictions with competitive MAE.

**Weaknesses:**

I find the writing subpar in general. Here are some detailed points:
1. The opening paragraphs used some awkward phrases.
2. The “Contributions” section lists the models and fail to emphasize their significance.
3. The PCA in Section IV A is introduced in the Experiments rather than the Methods. Since this method is used to select bilirubin, the authors should briefly preview the PCA rationale in the methodology and explicitly cite literature for bilirubin’s clinical importance.
4. Tables I and II report DT performance only. For completeness, include ML coverage and MAE alongside DT metrics so readers can directly compare them.
5. Key implementation details (e.g., regression algorithms, parameter settings, code availability) are omitted. A link to a GitHub repository or supplementary appendix would help readers reproduce and extend your work.

---

### Official Review · Reviewer_zinL · 2025-07-14
**This paper presents a comparative framework for predicting liver biomarker ranges using machine learning and physiologically-informed digital twin modeling.**

**Confidence:** 3
**Clarity Of Writing:** good
**Clinical Significance:** fair
**Methodological Novelty:** good
**Overall Rating:** 6
**Final Rating:** 7

**Experiments And Results:**

fair

**Questions For The Authors:**

How do the predicted bilirubin ranges translate to clinical decision-making thresholds, and how do they compare with established reference ranges used in clinical practice?

Why was a simple regression model chosen over more ML approaches like random forests?

What is the biological and clinical rationale for the specific physiological parameters and equations used in TWIN-SCAN's submodules?

**Strengths:**

Clinically relevant: Tackles a real-world issue in early-stage liver disease detection using non-invasive modeling approaches.

Dual-method comparison: Side-by-side evaluation of ML and DT approaches is novel and well-motivated, addressing a gap in prior literature.

Personalization: The inclusion of demographic variables (age, gender) in DT modeling increases biological plausibility and aligns well with clinical heterogeneity.

Comprehensive evaluation: Uses real-world datasets and interpretable metrics (coverage, MAE) to validate model performance.

Well-structured narrative: The methodology, analysis, and results are clearly presented with appropriate figures and tables.

**Summary Of The Paper:**

This paper addresses early-stage liver disease detection with clinical biomarker data. It compares two methods for estimating bilirubin levels: (1) a machine learning approach that uses cirrhosis-stage data and healthy references to predict values for underrepresented stages (NAFLD and fibrosis), and (2) TWIN-SCAN, a physiology-informed model simulating bilirubin conjugation pathways with three variants (Generic, Tier-1 Personalized by gender, and Tier-2 by gender and age). Validated with the Cirrhosis Patient Survival Prediction dataset, results indicate the ML approach has higher coverage accuracy, while TWIN-SCAN offers more physiologically plausible and narrower prediction ranges with lower mean absolute error.

**Weaknesses:**

Independent external validation on different clinical cohorts would strengthen the claims, since the NAFLD and fibrosis stages' ground truth are estimated.

Comparing this method with existing non-invasive liver assessment techniques or biomarker panels currently used in clinical practice could make the results more convincing.

The digital twin model, although physiologically motivated, seems to be based on simplified assumptions about bilirubin metabolism without detailed biochemical validation or comparison to established pharmacokinetic models. While bilirubin is an important clinical marker, the model does not include other key liver biomarkers, which could provide additional insights. The paper also does not explain how the predicted biomarker ranges would apply in a clinical setting.

It would be helpful to see the statistical significance of the reported improvements.

---

### Official Review · Reviewer_2Db8 · 2025-07-14
**Interesting topic but weak experiment results**

**Confidence:** 4
**Clarity Of Writing:** good
**Clinical Significance:** good
**Methodological Novelty:** good
**Overall Rating:** 3
**Final Rating:** 5

**Experiments And Results:**

fair

**Questions For The Authors:**

Could you give a brief description of the dataset you used in the paper? Provide some information like sample size, distribution of healthy population and cirrhosis population, distribution of bilirubin among the samples.

You optimized your digital twin model with personalized information including age and gender to yield better performance for individuals. So why did you use MAE between digital midrange and dataset mean? Mean absolute error between the regression results and ground truth can be more supportive. If you wanted to compare the results and the ground truth group-wise, maybe statistical analysis between the two distributions is a better choice.

Please give more details of the machine learning model you used. Many popular machine learning methods like random forest, support vector machine can do regression task. Could you give more details about the one you used? Different machine learning models also perform differently on the same task.

If these questions can be answered clearly and reasonably, I will increase my score to experiement and result.

**Strengths:**

The authors proposed non-invasive methods for early-stage liver diseases detection and creatively made full use of available data of healthy participants and participants with cirrhosis to address the problem.

The authors compared two popular kinds of models, machine learning based model and digital twin based model in early-stage liver diseases detection. It's valuable to see comparisons between methods based on totally different inference principles.

**Summary Of The Paper:**

The author compared two methods, a machine learning based model and a physiology-informed partial digital twin model, to estimate bilirubin conjugation for early-stage liver diseases detection.

**Weaknesses:**

In the introduction section, application of digital twin modeling in liver pathology analysis was not discussed.

Vague description of the proposed machine learning based method. Machine learning model is a type of models comprising of many algorithms, like linear regression, random forest, neural network, and so on, rather than a specific model.

Lack of description of the dataset. The size and distribution of dataset is critical for developing and evaluating models. And the information used as input or used for building mathematical model is also important. This information should be provided explicitly in the paper.

The evaluating results using coverage accuracy and mean absolute error between digital twin midrange and dataset mean (this is also confusing because the definition in the paper and the description in the caption of Table 2 are different) is not supportive enough for the conclusion. For example, three patients A, B, and C with ground truth 4, 5 and 6 can be predicted as values 10, 5, 1, with a 100% coverage accuracy and 0 MAE (as defined in the caption of Table 2), which seems perfect but actually not.

Some language mistakes like "Currently liver disease is a significant and one of the growing global health challenges", "physiolog-informed", "PCN", "principal component analysis (PCS)".

---

### Official Review · Reviewer_wvRP · 2025-07-17
**paper revision**

**Confidence:** 4
**Clarity Of Writing:** good
**Clinical Significance:** good
**Methodological Novelty:** great
**Overall Rating:** 6

**Experiments And Results:**

good

**Questions For The Authors:**

Non-Reproducible Methodology
and other concerns mentioned in weakness.

**Strengths:**

The paper is motivated by a well-defined clinical need non-invasive detection of liver disease. Its great Motivation and liked it.
The authors do a good job of contextualizing the problem and highlighting the limitations of existing data and methods.

The kind of comparison between a model and a physiology-informed simulation is a strong and interesting conceptual framework. The idea of using a partial, personalized digital twin to fill gaps in clinical data is promising

**Summary Of The Paper:**

Based on my understanding, paper propose a framework to estimate bilirubin levels for early-stage liver disease, where clinical data is often unavailable. They compare a standard ML extrapolation method with a new physiology-informed DT model TWIN-SCAN. This DT model simulates the bilirubin conjugation pathway and can be personalized using patient age and gender.
The idea is that the DT approach produces more clinically accurate biomarker estimates than the ML model, esp for the underrepresented NAFLD and fibrosis stages.

**Weaknesses:**

I feel like this paper and its method as Non-Reproducible Methodology

no mathematical formulation, governing equations, or parameterization.
Insufficient and Contradictory Experimental Validation

The results presented in Table II show that the "improved" personalized model has a significantly higher error (MAE) for Cirrhosis Stage 3 than the generic model (1.75 vs. 0.82). This doesnt seem is neither explained nor addressed. raise doubt on the entire validation process.

Weak Baseline Comparison
The ML model, simple model trained on data distribution, is used for an extrapolation task. it is not a strong or representative baseline.